# Tranexamic Acid in Hip Reconstructions in Children with Cerebral Palsy: A Double-Blind Randomized Controlled Clinical Trial

**DOI:** 10.3390/children10121931

**Published:** 2023-12-15

**Authors:** Alexandre Zuccon, Paulo Rogério Cardozo Kanaji, Dávia Serafini Barcellos, Saulo Zabulon, Ageu de Oliveira Saraiva, Thaila Andressa Yoshi de Freitas

**Affiliations:** Disabled Children’s Care Association of São Paulo (AACD-SP), São Paulo 04027-000, Brazil; pkanaji@aacd.org.br (P.R.C.K.); davilaserafini@hotmail.com (D.S.B.); saulozabulon@hotmail.com (S.Z.); ageu.saraiva27@gmail.com (A.d.O.S.); thaila.yoshii.freitas@gmail.com (T.A.Y.d.F.)

**Keywords:** cerebral palsy, hip dislocation, hip reconstruction, bleeding, tranexamic acid

## Abstract

Surgical treatment is indicated for hip dislocation in patients with cerebral palsy (CP), but it requires care due to the state of nutrition and associated clinical comorbidities. The use of resources that minimize blood loss and the need for blood transfusions are essential to avoid complications. Tranexamic acid (TXA) has been highlighted for orthopedic surgeries to control intraoperative bleeding; however, there is a lack of large studies for its use in hip surgeries in patients with CP. This study aims to evaluate the efficacy and safety of tranexamic acid to reduce bleeding in pediatric patients with cerebral palsy undergoing surgical treatment for hip instability. A sample of 31 patients with CP who underwent surgical treatment for hip dislocation (hip adductor stretching, varization osteotomy of the proximal femur and acetabuloplasty using the Dega technique) was randomly divided into groups: control (*n* = 10) and TXA (*n* = 21). Preoperative and 24 h hemoglobin concentrations, the length of hospital stay (LHS), and intraoperative bleeding (IB) were analyzed. TXA significantly reduced the IB (*p* = 0.02). The variance in hemoglobin concentration was lower for the TXA group, but without statistical significance (*p* = 0.06). There was no difference in LHS. Also, no statistical difference was observed for the number of transfusions (*p* = 0.08). The findings provide evidence of the effectiveness of TXA in decreasing intraoperative bleeding and its safety for use in pediatric patients with cerebral palsy.

## 1. Introduction

Hip instability is a frequent complication in children with cerebral palsy (CP), especially in those with the spastic form and in those who cannot walk. The evolution of hip instability to subluxation and dislocation is currently an issue in orthopedic practice. The treatment of hip subluxation or dislocation is surgical; therefore, perioperative blood loss is a frequent cause of complication [1,2].

Excessive bleeding during and after surgery can interfere with the treatment outcome. Multiple strategies have been studied to reduce blood loss and the need for transfusion, such as the use of controlled hypotensive anesthesia, aspiration and blood recovery systems, cryoprecipitate plasma and antifibrinolytic drugs, with variable risks and costs [3]. Many of these methods are available for children, but efficiency will depend on the patient’s age and the type of surgery. Complication control related to blood loss can take place in three areas: red blood cell mass preoperative increase, blood loss perioperative decrease and transfusion practice optimization; the combination of the three is the best strategy, depending on the individual risk evaluation and the surgery involved. When large blood loss is expected, greater are the benefits of blood conservation strategies. The infusion of antifibrinolytic drugs during the surgical procedure has been promisingly reported to reduce perioperative bleeding [3,4,5,6,7].

Tranexamic acid (TXA) is a rapidly absorbed synthetic antifibrinolytic drug [8], capable of inhibiting fibrinolysis through the formation of a reversible complex with plasminogen and plasmin, preserving fibrin clots and partially blocking platelet aggregation, and acting as a hemostatic adjuvant in the prevention of hemorrhages [9]. Its use may be indicated for orthopedic and cardiovascular surgeries, epistaxis, gastrointestinal tract hemorrhages and obstetric [10] and pediatric [11,12,13] emergencies.

In orthopedic medical practice, this medication is already used in major surgical procedures, with support in the literature for hip arthroplasties in adults [14,15], spinal surgical procedures [16,17,18] and post-trauma bleeding control in pediatric patients [10]. However, there are few studies proving the efficiency of TXA in children with CP undergoing hip surgery. Cerebral palsy and its associated comorbidities predispose one to pre- and postoperative complications, the need for blood transfusions and prolonged hospitalization periods [1].

Recent studies on the effect of TXA on hip reconstruction surgeries in children with CP have demonstrated variable results. Regarding the need for blood transfusion, length of hospital stay, intraoperative bleeding and drop in hemoglobin, improvements without statistical relevance have already been reported, some with limitations due to lack of transfusion criteria or lack of data in medical records (retrospective studies) [19].

Others did not define an improvement in the levels of hemoglobin drop or need for transfusion, but without randomization methods, more severe patients (worse nutritional status and major surgeries) were grouped in the group that received TXA, which may have influenced these results [20].

The objective of this study was to evaluate the efficacy and safety of tranexamic acid by evaluating blood transfusions, intra- and postoperative blood loss, hemoglobin variance, and the length of stay in UT in patients with cerebral palsy undergoing surgical reconstruction of the hip, using a method of double-blind randomization and standardized hemoglobin collection before and after the procedure. 

The aim of this study was to evaluate the efficacy and safety of tranexamic acid for reducing bleeding in patients with cerebral palsy undergoing surgical hip reconstruction.

## 2. Materials and Methods

### 2.1. Study Design

This is a randomized, double-blind clinical trial, which included thirty-one pediatric patients with cerebral palsy, with subluxation or dislocation of the hip. All of them underwent hip reconstruction through hip adductor stretching, varization osteotomy of the proximal femur and acetabuloplasty using the Dega technique, from June 2018 to November 2019 in our hospital. The Research Ethics Committee approved this research, with protocol number 88131318.2.0000.0085.

### 2.2. Inclusion and Exclusion Criteria

The following were considered as inclusion criteria in the study: minors under 18 years of age; patients diagnosed with Cerebral Palsy; preoperative hemoglobin greater than 10 g/dL; and unilateral hip surgical reconstruction. Patients who underwent simultaneous bone surgery in another anatomical region, patients with bleeding disorders or thrombophilia, patients with chronic kidney disease (creatinine clearance less than 60 mL/min·m^2^) and those with refusal to consent to participate were excluded from the study. Patients who received intraoperative transfusions were excluded from the hemoglobin concentration variance analysis, as transfusion distorts this parameter.

### 2.3. Methodological Criteria

The randomization process occurred in preoperative consultations through the random distribution of research participants into two groups: Tranexamic Acid Group and control group (placebo). It was ensured that it was impossible to establish the advantages of one procedure over another, through a literature review and observational methods involving human beings.

The intergroup distribution was generated by Random Number Generator software (Randomizer V1.0.7, UXAPPS LTD, Android^®^ application), which allowed the configuration of truly comparable groups, so that each patient had the same probability of belonging to one of the groups (experimental or control). Thus, any differences in the occurrence of the outcome between the groups can be attributed to the intervention.

The surgical team, the patient, and the immediate postoperative management team (intensivists and interns) were blinded to the patient’s study group allocation. Only the anesthetist and his assistant knew which group the patient was allocated to and were responsible for administering TXA or placebo (0.9% saline solution).

Patients allocated to the TXA group received the antifibrinolytic agent at a dose of 10 mg/kg in an intravenous bolus, single dose, during the anesthetic act, while the control group received 0.9% saline solution as a placebo. The volunteers in the placebo group were previously informed about the use of saline solution during surgery, which is a safe resource, with no contraindications and already routinely used in orthopedic procedures.

The hemoglobin reduction rate (24 h pre-operative hemoglobin value − 24 h postoperative hemoglobin value), the number of days spent in the ICU (intensive care unit), the total length of hospital internment, the volume of intraoperative bleeding (estimated by weighing the surgical compresses used during the procedure, in other words, weight of surgical compresses wet with blood − weight of dry surgical compresses) and the in-hospital complications were recorded and documented for comparison between the two groups.

### 2.4. Risks and Benefits

The benefits established by the study were decreased intraoperative and immediate postoperative bleeding.

The risk identified was the possibility of an adverse reaction to tranexamic acid.

### 2.5. Statistical Analysis

The non-parametric Kruskal–Wallis test was used using the SPSS software version 20, Minitab 16, and the t-Student test, with a statistical significance index of *p* < 0.05. 

## 3. Results

Our sample consisted of thirty-one patients who underwent unilateral proximal femoral varus osteotomy associated with acetabuloplasty using the Dega technique. In the total group, twenty were male and 11 were female; the average age was 124 months (minimum 64 months and maximum 216 months). All were diagnosed with Cerebral Palsy: twenty-three spastic quadriplegic, 4 mixed quadriplegic and 4 spastic diplegic. According to the Gross Motor Function Classification Scale (GMFCS), one was GMFCS 1, ten was GMFCS 4 and twenty was GMFCS 5.

Twenty-one patients received a dose of TXA 30 min before the procedure: the patients’ average age and weight were 10.27 years and 22.62 kg. Ten patients did not receive TXA: the average age and weight was 10.69 years and 31.8 kg. Only weight represented differences; ages were similar (Table 1).

### 3.1. Blood Loss

The mean bleeding per patient analyzed intraoperatively was 194 mL for the TXA group and 313 mL for the control group (*p* = 0.002). We found that three children in the control group and six in the intervention group were transfused; even with a lower blood loss in the group that received the acid, there was a greater number of transfused patients, without statistical significance (*p* = 0.08). This fact demonstrates that the criteria for blood transfusion in patients with cerebral palsy depend on several variables in addition to intraoperative blood loss, such as previous nutritional status. The average surgery time was similar between the groups, being 165 min (TXA) and 168 min (control).

Postoperative bleeding from postoperative drains was quantified, but we believe they are not valuable for this analysis due to the lack of a protocol or standardization criteria for use (some surgeons used it, and others did not at their own discretion). These values are in Table 2. 

### 3.2. Drop in Hemoglobin Concentration

The mean rate of reduction in hemoglobin (Hb) was 2.38 dL for the TXA group and 3.17 for the control group (*p* = 0.07); although an improvement was observed in this parameter, there was no statistical significance (Table 3). 

### 3.3. Complications

For postoperative complications, two cases of pneumonia were identified, one patient in each group, and both were submitted to mechanical ventilation and used anticonvulsant as a common aspect. No bleeding disorders or even adverse effects were observed in any of the groups.

### 3.4. Hospital Internment

The average length of stay required in the ICU was 3.2 days for the control group and 1.43 days for the TXA group. The total length of hospital stay was 5.9 days for the control group and 3.71 days for the TXA group; although the use of TXA demonstrated a reduction in hospital stays, there was no statistical relevance (Table 4).

Finally, we used the Minitab technique to calculate the power and sample size, based on the HB preoperative statistics (Table 5).

The difference statistic was 1.3 times the value of the confidence interval at 95% statistical confidence. Thus, the power of the sample with 31 cases was 72.2% compared to 72.1% in the sample with 22 cases.

## 4. Discussion

Patients with cerebral palsy (CP) are at risk of developing several orthopedic comorbidities, including hip instability and progression to subluxation and dislocation, a frequent problem occurring in 3% to 47% of patients, varying with the severity of neurological involvement. The incidence of hip subluxation and dislocation is also linked to the patient’s ability to walk, reaching 89% in non-ambulatory patients. Patients present with subluxation or dislocation at around 7 years of age on average [21]. Hip subluxation develops in response to muscular imbalance: hip adductors and flexors more spastic and shortened in antagonism to the more weakened extensors and abductors. Consequently, lateralization and proximal migration of the femoral head in relation to the acetabulum occurs. This mechanism is completely different from developmental dysplasia of the hip, in which soft tissue laxity and acetabular changes, often present at birth, lead to hip instability [22].

The goal of treatment is a painless hip that allows proper positioning and joint stability. In a more significant hip instability scenario, the most frequently performed procedure is the proximal femur external rotation osteotomy and varization, associated or not with pelvic osteotomy. Patients with CP require greater pre- and postoperative care because they present the involvement of systems other than the neuromuscular, in addition to complications such as seizures, dysphagia or reactive airway disease. For this reason, a systematic method of assessment, planning, and integration of elements for proactive intervention helps more efficient hospitalizations, greater satisfaction and better results [23].

Intraoperative blood loss is a predictive factor for hypotension, coagulopathy, anemia, inadequate tissue perfusion, renal failure, cardiac disorders, and the need for blood transfusion. However, blood transfusion in children has a higher incidence of adverse effects with worse outcomes when compared to adults, so we must avoid both situations. Patients with CP are more vulnerable to blood loss due to the use of anti-convulsant medications, exhaustion of clotting capacity and frequent severe nutritional deficiencies [3]. However, data regarding the application of antifibrinolytics in hip surgical procedures in pediatric patients with CP are scarce, while there are numerous promising pieces of evidence regarding their use in arthroplasties [14,15], spinal surgical procedures [16,17,18] and the control of post-trauma bleeding in pediatric patents [10].

In a data analysis from the American College of Surgeons National Surgical Quality Improvement Program Pediatric (ACS-NSQIP Pediatric), identified 1184 surgical procedures for hip dysplasia treatment between 2012 and 2013. This series was distributed into three groups, the first being composed of patients without any associated comorbidity (*n* = 451), the second with individuals with associated comorbidities (*n* = 216) and the third group consisting of patients with neuromotor disorders (*n* = 517) such as cerebral palsy, tetraplegia, paraplegia, and hemiplegia. The incidence of blood transfusions was higher in the third group, present in 31.3% of these patients [24].

In a retrospective analysis of orthopedic spinal surgeries, which included 272 patients with cerebral palsy, sought to identify the main causes of hospital complications after orthopedic spinal procedures. Blood loss corresponding to one volume (bag) was identified in 39.7%, whose need for blood transfusion is correlated with the complexity of the procedure and incidence of postoperative complications [7]. In the present study, only two cases of post-surgical complications were identified: one patient in each group presented with pneumonia. Both patients had a history of the continuous use of anticonvulsant drugs and had been subjected to mechanical ventilation. The use of anticonvulsants poses an additional risk for intraoperative blood volume output and the incidence of complications [23].

Varization and external rotation osteotomies of the proximal femur associated with acetabuloplasty are considered major surgical procedures for patients with CP and have been recommended as an effective treatment alternative [22]. Canavese et al. (2014) described pelvic osteotomy performed using a percutaneous surgical technique in patients with severe CP and acetabular dysplasia. However, the authors did not provide data regarding blood loss resulting from the surgical procedure [25].

Excessive bleeding during and after surgery can interfere with the treatment outcome. Multiple strategies have been studied to reduce blood loss and the need for transfusion, such as controlled hypotensive anesthesia, blood aspiration and recovery systems, cryoprecipitate plasma and antifibrinolytic drugs, with variable risks and costs [3]. Many of these methods are available for children, but effectiveness will depend on the patient’s age and the type of surgery. The complication control related to blood loss can take place in three areas: red blood cell mass preoperative increase, blood loss perioperative decrease and transfusion practices optimization; the combination of the three is the best strategy, depending on the individual risk evaluation and the surgery involved. When large blood loss is expected, greater are the benefits of blood conservation strategies. The infusion of antifibrinolytic drugs during the surgical procedure has been promisingly reported to reduce perioperative bleeding [3,4,5,6,7]. However, the heterogeneity of studies using antifibrinolytics and the lack of scientific evidence in patients with hip dislocation and subluxation, associated with cerebral palsy, limit the safe use of this resource in clinical practice.

Tranexamic Acid is an option that has been gaining prominence; TXA is a rapidly absorbed synthetic antifibrinolytic drug [8], capable of inhibiting fibrinolysis through the formation of a reversible complex with plasminogen and plasmin, preserving fibrin clots and partially blocking platelet aggregation [9], with use indicated for orthopedic and cardiovascular surgeries, epistaxis, gastrointestinal tract bleeding and obstetric [10] and pediatric [11,13] emergencies.

In Brazil, following the national health surveillance agency (ANVISA) recommendations, the dose indicated for the pediatric community is 10 mg/kg, and it can be repeated two to three times in a period of 24 h [26]. The maximum recommended daily dose, for all indications, is 3 g/day. However, in some cases and under supervision, it can be increased up to 4.5 g/day. In some surgeries, such as heart surgeries, liver transplants and major orthopedic surgeries, the maximum daily dose of tranexamic acid may vary according to the patient’s needs and individual professional experience, thus remaining at medical discretion [26].

For this study, it was chosen to follow the standards described in the Brazilian leaflet, for patient safety. This dose demonstrated efficiency in significantly reducing intraoperative bleeding (*p* = 0.02), even though it was lower than the dose established by the Royal College of Pediatrics and Child Health: 1 g during the surgical procedure, followed by another 1 g over 8 h for children with age under 12 years [27].

Regarding intraoperative bleeding, Dhawale et al. (2012) compared the results between tranexamic acid, epsilon aminocaproic acid and the control [3]. Although no significant difference was identified for cellular markers, the drugs contributed to a bleeding reduction. The use of these resources also allowed an average reduction of 4 days of hospitalization as well as the time of mechanical ventilation, evaluated at 2.4 days, compared to 3.8 for patients without intervention. In another retrospective study, positive evidence was also identified for the use of epsilon aminocaproic acid during spine surgeries in pediatric patients, with a significant blood loss reduction. However, this drug was discontinued in 2007 and replaced by tranexamic acid, with a similar action mechanism, but with antifibrinolytic capacity up to ten times more potent [6]

A study presented contradictory results after applying the TXA in 14 of 51 hip reconstructions for patients with CP. The researchers describe that in the group of patients who were assigned the antifibrinolytic, the rate of blood transfusions was higher (>6.8%; average 42.9% of patients) and they remained hospitalized for longer (>0.54 day; average 8.17 days). Comparable results have also been reported for blood loss. However, that study considered, as a factor for inclusion in the TXA experimental group, only patients with severe CP, and a considerable risk of bleeding due to the history of use of anticonvulsants, in addition to the need for a bilateral surgical procedure. Even though the drug was administered, such factors may have significantly influenced intraoperative blood loss and may have contributed to doubtful results regarding the use of TXA [1].

Zekcer A et al. (2017) compared in a randomized clinical trial the use of topical and intravenous tranexamic acid in total knee arthroplasty. They demonstrated a reduction in blood loss and the need for transfusion. Topical use showed comparable results to intravenous use, but less side effects [28].

The retrospective study by Rocha et al. (2015), in which the medical records of 40 patients undergoing posterior thoracolumbar arthrodesis were analyzed, demonstrated that TXA was effective in reducing perioperative bleeding, as demonstrated in other studies, but they were unable to establish the correlation between its use and reducing the need for blood transfusion [16].

Almeida et al. (2017) sought to evaluate in a clinical trial the effectiveness of tranexamic acid in reducing bleeding in patients undergoing total knee arthroplasty. When comparing the groups, a statistical difference (*p* < 0.05) was observed in the following parameters: reduction in hemoglobin, reduction in hematocrit, estimated blood loss and drain output. All values were lower in the tranexamic acid group. Only patients in the placebo group required blood transfusion [29].

Benjamin J. et al. in 2020 conducted a retrospective study with 166 patients and observed that the use of TXA during hip reconstruction surgery in children with neuromuscular disease significantly reduced the percentage loss of estimated blood volume and the postoperative transfusion rate [20].

In our analysis, we observed that the use of tranexamic acid significantly reduced intraoperative bleeding in patients with CP (*p* = 0.02). To maintain hemoglobin 24 h after surgery, a smaller variation in Hb was observed for the group undergoing intervention, but without statistical significance (*p* = 0.06). Regarding the length of stay (*p* = 0.08) and ICU time (*p* = 0.35), there was no difference between the groups, unlike previous studies that demonstrated a relevant decrease in this aspect. We found that three children in the control group and six in the intervention group were transfused; even though there was less blood loss in the group that received the acid, there was a greater number of transfusions, without statistical significance (*p* = 0.08). This fact demonstrates that the criteria for blood transfusion in patients with cerebral palsy depend on several variables in addition to intraoperative blood loss, such as previous nutritional status. For postoperative complications, two cases of pneumonia were identified, one patient in each group, and both had undergone mechanical ventilation and used anticonvulsants as a common aspect. The medication proved to be safe for use in pediatric patients with cerebral palsy, as no bleeding disorders or adverse effects were observed in any patient. The dosimetry used was a 10 mg/kg intravenous bolus, single dose, during anesthetic induction.

## 5. Conclusions

Tranexamic acid demonstrated efficacy in significantly reducing intraoperative bleeding, a fact that could suggest a reduction in complications related to blood loss. There was a smaller drop in hemoglobin concentration 24 h after the surgical procedure in patients who received tranexamic acid, although this was not statistically significant. We found no significant differences between the TXA and control for the length of hospitalization and ICU, as well as number of transfusions. The use of the drug showed comparable results regarding hospital complications in relation to the control group. TXA proved to be safe for use in pediatric patients with cerebral palsy, as no patient experienced adverse effects related to the drug. We hope that interest in studying methods that reduce the impact of major surgeries that children with cerebral palsy undergo, including tranexamic acid, will increase. We know that these patients already have a low weight and altered nutritional status and that rapid recovery is important for rehabilitation and an improved quality of life.

## Figures and Tables

**Table 1 children-10-01931-t001:** Demographics for TXA versus control groups.

		Average	Median	SD	CV	Min	Max	N	CI	*p*
Weight	TXA	22.62	22.00	7.26	32%	9.0	37.0	21	3.11	0.014
	Control	31.80	30.50	12.42	39%	18.0	50.0	10	7.70	
Age	TXA	10.27	10.50	2.57	25%	5.3	14.7	21	1.10	0.721
	Control	10.69	9.71	3.92	37%	5.4	18.0	10	2.43	

**Table 2 children-10-01931-t002:** Bleeding for TXA versus control groups.

		Average	Median	SD	CV	Min	Max	N	CI	*p*
Bleeding (intraoperative)	TXA	194	180	90	46%	95	470	21	38	0.002
	Control	313	285	92	29%	220	530	10	57	
Bleeding (postoperative)	TXA	96.8	105.5	42.2	44%	30.0	150.0	10	26.2	0.498
	Control	115.0	120.0	70.3	61%	20.0	265.0	9	45.9	

**Table 3 children-10-01931-t003:** Hb variation for TXA versus control groups.

		Average	Median	SD	CV	Min	Max	N	CI	*p*
Hb Pre	TXA	12.59	12.40	1.14	9%	10.7	14.7	21	0.49	0.365
	Control	12.95	12.95	0.68	5%	12.1	14.1	10	0.42	
Hb Post	TXA	10.21	10.10	1.19	12%	8.3	13.1	21	0.51	0.310
	Control	9.78	9.70	0.80	8%	8.9	11.1	10	0.49	
Hb Variation	TXA	−2.38	−2.30	1.15	−48%	−4.9	0.0	21	0.49	0.070
	Control	−3.17	−3.15	0.95	−30%	−4.3	−1.1	10	0.59	
Hb Efficiency	TXA	−18.7%	−18.8%	8.3%	−45%	−37.1%	0.0%	21	3.6%	0.074
	Control	−24.3%	−25.2%	6.9%	−28%	−31.8%	−9.0%	10	4.3%	

**Table 4 children-10-01931-t004:** Length of hospital stay for TXA versus control groups.

		Average	Median	SD	CV	Min	Max	N	CI	*p*
Days at UCI	TXA	1.43	1.00	0.81	57%	1.0	4.0	21	0.35	0.098
	Control	3.20	1.00	4.69	146%	1.0	16.0	10	2.90	
Total days hospitalization	TXA	3.71	3.00	1.35	36%	2.0	7.0	21	0.58	0.049
	Control	5.90	4.00	4.56	77%	3.0	18.0	10	2.82	

**Table 5 children-10-01931-t005:** Statistics for calculating sample power.

Hb	All Sample	Without Transfusion Sample
Average	12.71	12.73
Standard deviation	1.014	1.124
N	31	22
Difference	0.464	0.610
Power	72.2%	72.1%

## Data Availability

The data presented in this study are available in article.

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
