# Peer review of "Tranexamic Acid in Hip Reconstructions in Children with Cerebral Palsy: A Double-Blind Randomized Controlled Clinical Trial"

_children, 2023, doi:10.3390/children10121931_

Round 1

Reviewer 1 Report

Comments and Suggestions for Authors

Dear authors,

thanks for submitting your original research to Children Journal. Based on my experience it's an important and novel study. Well done.

There are a couple of questions:

1.) I am missing a power calculation. How was the sample size calculated?

2.) Who did the randomisation? Was this person involved in the study?

3.) I kindly ask you to provide details of all enrolled subjects, such as demographics, age, sex, GMFCS levels, type of surgery and duration in hours and minutes of the surgical procedure.

In my opinion the paper written by RUTZ et al. in 2015 , Am JBJS, is the gold standard paper in regards of surgical hip recon in children with CP an should be referred as well.

4.) When was this first postoperative Hb measurement performed exactly ? Always 24 hours after the end of the surgery? How did you deal with morning and afternoon cases? Did you consider a second Hb value e.g. after 3 days after the surgery?

5.) How was blood loss intraoperatively assessed? Please describe in detail.

6.) Stats: Lines 108 and 109 don't sound accurate to me for such a complex and valuable RCT. I recommend review by a professional expert with a stats background.

7.) Not sure if all your conclusions are supported by your data. Mainly I recommend a new followup point at about 3 months in addition with a new clinical assessment and Hb check.

With best wishes, the reviewer

Comments on the Quality of English Language

Needs English check, please.

Author Response

1.) I am missing a power calculation. How was the sample size calculated? Presented in table 2,3 and 4

2.) Who did the randomisation? Was this person involved in the study? Methodology: an independent assistant outside the surgical team using an Android randomization application

3.) I kindly ask you to provide details of all enrolled subjects, such as demographics, age, sex, GMFCS levels, type of surgery and duration in hours and minutes of the surgical procedure. Our study did not use these variables, the ones used (ICU time, time in hospital, blood loss and hemoglobin dosage) are better described in table 1

4.) When was this first postoperative Hb measurement performed exactly ? Always 24 hours after the end of the surgery? How did you deal with morning and afternoon cases? Did you consider a second Hb value e.g. after 3 days after the surgery? Methodology: 24-h pre-operative and 24-h post-operative Hb

5.) How was blood loss intraoperatively assessed? Please describe in detail. Methodology :weighing surgical pads

6.) Stats: Lines 108 and 109 don't sound accurate to me for such a complex and valuable RCT. I recommend review by a professional expert with a stats background. Reviewed

7.) Not sure if all your conclusions are supported by your data. Mainly I recommend a new followup point at about 3 months in addition with a new clinical assessment and Hb check. Longer follow-ups and more detailed variables may be the subject of future study, which we are currently unable to carry out.

Reviewer 2 Report

Comments and Suggestions for Authors An interesting article on a previously poorly described topic. The article evaluates TRANEXAMIC ACID IN HIP RECONSTRUCTIONS IN CHILDREN WITH CEREBRAL PALSY: A DOUBLE BLIND RANDOMIZED CONTROLLED CLINICAL TRIAL. The aim of the study was to evaluate TRANEXAMIC ACID for intraoperative bleeding. Abstract: Well written. Introduction: Well-written, broad review of the literature. A clearly stated purpose of the study. Please add a research hypothesis. Material and methods: Well described. The study group and research methods are presented. Please describe in more detail the inclusion and exclusion criteria for the study. Please provide details of the age and gender of patients in both groups. Please describe why there were 31 people in the study. Results: Well-presented, nice, legible and of good resolution tables and figures. Discussion: Well presented, broad review of the literature. references: good overview.

Author Response

Improved methodology and statistical study data in tables.

Round 2

Reviewer 1 Report

Comments and Suggestions for Authors

Dear authors,

thanks for R1 new version. Your manuscript still need a lot of work and re-writing before it could be considered for acceptance for publications. My main concerns are still:

1. I am missing a power calculation and sample size calculation. I recommend that you consult advice from professional stats experts.

2. I am not happy with the description of the estimate of the blood loss. What was weighed? Please describe any details.

3. Randomization: did the surgical team and the anaesthetists know? How was TXA administered?  You mention that your study is DOUBLE blinded. Please provide details.

Thanks for your understanding.

Best wishes, the reviewer

Comments on the Quality of English Language

Needs re-writing and English check.

Author Response

1. I am missing a power calculation and sample size calculation. I recommend that you consult advice from professional stats experts.= line163 and table 5

2. I am not happy with the description of the estimate of the blood loss. What was weighed? Please describe any details. = "the volume of intraoperative bleeding (estimated by weighing the surgical compresses used during the procedure, in other words, weight of surgical compresses wet with blood - weight of dry surgical compresses" line 101 - 103

3. Randomization: did the surgical team and the anaesthetists know? How was TXA administered?  You mention that your study is DOUBLE blinded. Please provide details. = "The surgical team, the patient, and the immediate postoperative management team (intensivists and interns) were blinded to the patient's study group allocation. Only the anesthetist and his assistant knew which group the patient was allocated to and were responsible for administering TXA or placebo (0.9% saline solution)" line 90-03